# Effects of Celastrol-Enriched Peanuts on Metabolic Health and the Development of Atherosclerosis

**DOI:** 10.3390/nu17091418

**Published:** 2025-04-23

**Authors:** Jiaxin Shi, Yitong Cheng, Chenxuan Wang, Min Liu, Mingxuan Qu, Shuaishuai Zhou, Leon Chen, Xiaohao Li, Junjie Luo, Yongting Luo, Chao Luo, Peng An

**Affiliations:** 1Department of Nutrition and Health, Key Laboratory of Precision Nutrition and Food Quality, China Agricultural University, Beijing 100193, China; s_jx0717@163.com (J.S.); ccc666678@126.com (Y.C.); wangchenxuan0609@163.com (C.W.); luojj@cau.edu.cn (J.L.); 2Guangdong Agricultural Antibiotic Reduction and Replacement Technology Promotion Association, Shenzhen 518114, China

**Keywords:** celastrol-enriched peanut, atherosclerosis, blood lipid, anti-inflammatory, intestinal flora

## Abstract

Background: Celastrol, a pentacyclic triterpenoid active component isolated from the root bark of the traditional medicinal plant Tripterygium wilfordii, displays significant anti-inflammatory, antioxidant, and immunomodulatory properties. However, its clinical application remains limited due to inadequate bioavailability. Methods: Regarding these issues, we innovatively developed a novel peanut cultivar (cel-peanut) enriched with celastrol through distant hybridization combined with metabolomics screening. Guided by the research concept of “natural anti-inflammatory diets for metabolic disease management”, we established a high-fat diet-induced *ApoE*^−/−^ atherosclerotic mouse model to systematically evaluate the anti-atherosclerosis effects and mechanisms of cel-peanut. Results: Our results revealed that cel-peanut significantly reduced serum levels of triglycerides (TGs) and increased high-density lipoprotein cholesterol (HDL-C). Concurrently, cel-peanut markedly decreased the atherosclerotic lesion area and enhanced collagen content within plaques. Mechanistic investigations demonstrated that cel-peanut reduced serum malondialdehyde (MDA) levels and suppressed the concentration of pro-inflammatory cytokine IL-6 in atherosclerotic lesions. Furthermore, cel-peanut promoted intestinal health by modulating the composition and functionality of gut microbiota, thereby attenuating atherosclerosis progression. Conclusions: Overall, these findings indicate that cel-peanut exerts therapeutic effects against atherosclerosis through its anti-inflammatory, antioxidant, and gut microbiota-modulating properties. This study proposes a novel nutritional intervention strategy for atherosclerosis and provides a promising adjuvant strategy for clinical atherosclerosis treatment.

## 1. Introduction

Cardiovascular diseases are the leading cause of mortality, accounting for nearly 17.9 million lives each year [1]. Atherosclerosis (AS) is a chronic inflammatory disease that serves as the main pathological basis of most cardiovascular diseases [2]. Various risk factors, including obesity, genetic history, and age [3], have been considered as risk factors for atherosclerosis. In addition, there are additional non-traditional risk factors for atherosclerosis, such as poor diet quality, sedentary lifestyle, ambient air pollution, and psycho-social stress [4].

The pathogenesis of AS involves the following: increased endothelial permeability and initiating inflammation caused by minor damage, allowing for low-density lipoprotein infiltration and oxidative modification in the arterial wall; monocyte recruitment and transformation into macrophages and foam cells; platelet adhesion and growth factor release; migration and proliferation of medial smooth muscle cells into the intima, synthesizing the extracellular matrix (collagen and proteoglycans); and progressive lipid deposition in macrophages and smooth muscle cells. This chronic and complex process involves a continuous vicious cycle of lipid accumulation, inflammatory responses, and cellular proliferation, ultimately forming lipid-rich cores and fibrotic plaques [5]. Therefore, targeting inflammation could be an effective therapeutical strategy to prevent the development of atherosclerosis. Existing treatment strategies targeting inflammation include inhibiting cytokines, manipulating adaptive immunity, and promoting pro-resolution mechanisms [6]. Additionally, an increase in the intake of plant-based foods (whole grains, fruits, vegetables, legumes, and nuts) is associated with a reduced risk of atherosclerosis. Similarly, substituting butter and other animal/tropical fats with oils rich in unsaturated fats can benefit cardiovascular health. This type of diet affects the entire metabolism [7,8].

Natural products are an essential source for discovering and developing new compounds for the treatment of cardiovascular diseases [9,10,11]. Celastrol, a pentacyclic triterpenoid isolated from the roots of *Tripterygium wilfordii*, has shown great potential in the treatment of a wide variety of diseases and received considerable attention in recent years [12]. A large number of studies have shown that celastrol has a protective effect on obesity, diabetes, silicosis, and other diseases [13]. Importantly, growing evidence suggests celastrol could be a promising agent against atherosclerosis [14,15]. However, further clinical application of celastrol is still hampered by its limited hydrolysis, low oral bioavailability, and poor tolerance in vivo [16]. In addition, celastrol can have cytotoxic, hepatocyte, and even neurotic effects at high concentrations or in the case of prolonged exposure [17]. Thus, it may be useful to explore a more effective and safer strategy for the treatment of atherosclerosis through the proper utilization of celastrol.

Recently, advanced drug delivery systems based on nanotechnology were reported to deliver celastrol in the treatment of cancer [18] and inflammatory diseases [19]. However, shortcomings still exist in these studies, such as material instability, immunogenicity, and systemic toxicity [20,21]. Linoleic acid, lecithin, high-quality protein, minerals, vitamins, and fiber that exist within peanut are considered to be good carriers for fat-soluble natural compounds, which may effectively alleviate the toxicity produced by natural compounds via changing their pharmacokinetic properties [7,22,23,24].

This variety of peanut that rich in celastrol was screened and obtained through breeding techniques combining distant hybridization and metabolomics analysis. Considering the great potential of celastrol in the treatment of inflammation-associated diseases, the efficacy of celastrol-enriched peanut on atherosclerosis was investigated in a mouse model of atherosclerosis (*ApoE*^−/−^ mice) fed a 10-week high-fat diet.

## 2. Materials and Methods

### 2.1. Celastrol-Enriched Peanut

The Paraguay Mountain Small Peanut is a small-fruited, black-skinned peanut variety characterized by thin shells, small yet plump seeds, and excellent flavor. However, this plant exhibits a relatively tall stature, long growth cycles, late maturation, limited branching, and low yields, making it unsuitable for large-scale cultivation in China. In contrast, the Yangjing variety, a local germplasm from Wuzhi Mountain in Hainan Province, is renowned for its early maturity, abundant branching, and relatively high yields. This hybrid cultivar was developed through distant hybridization combining the high-yield Paraguay Mountain Small Peanut (maternal parent) with the Yangjing genotype (paternal parent). By integrating metabolomic analysis with distant hybridization, the secondary metabolic pathways of the plant were activated, achieving enrichment of celastrol while reducing its potential toxicity. This process preserved the superior flavor characteristics while maintaining stable high-yield traits. The combination of traditional hybrid breeding with modern metabolic pathway analysis has enabled synchronous optimization of phytochemical composition, sensory quality, and agronomic performance in this improved peanut variety.

Three types of animal feed were used in this experiment: (1) a high-fat purified diet (H10141, purchased from Beijing HFK Bio-Technology Co., Ltd., Beijing, China) with a fat–energy ratio of 41%, supplemented with 1.5% cholesterol, was used to induce high-fat and atherosclerosis models in mice; (2) a high-fat purified diet mixed with 20% regular peanuts per kilogram; and (3) a high-fat purified diet mixed with 20% celastrol-enriched peanuts per kilogram. The common peanuts (Luhua 11, celastrol content: 0.57 μg/kg) and celastrol-enriched peanuts (celastrol content: 1029.21 μg/kg) used in this study were provided by Hainan Misheng Biotechnology Co., Ltd. (Sanya, China).

### 2.2. Animals

Male apolipoprotein E-deficient mice (*ApoE*^−/−^, C57BL/6J background), aged 6 weeks, were purchased from Beijing Vital River Laboratory Animal Technology Co., Ltd. (Beijing, China). The mice were housed under specific pathogen-free conditions, with a temperature range of 20 to 26 °C, humidity of 40–70%, pressure of 45 Pa, illumination of 15–20 Lux, and 12 h light/dark cycle. All the experimental animals were fed a high-fat diet (HFD) for 10 weeks to induce obesity and atherosclerosis. The mice were randomly divided into three groups (6 mice per group): the blank high-fat diet group (HFD group), the high-fat diet with regular peanut control group (HFD-peanut group), and the high-fat diet with celastrol-enriched peanut intervention group (HFD-Cel-peanut group). Each group was fed their corresponding diets accordingly. The intervention lasted for 8 weeks, during which the mice had ad libitum access to food and water. Weekly records of body weight and feed consumption were kept throughout the intervention period. All the mice were sacrificed following intraperitoneal administration of 1.25% tribromoethyl alcohol for anesthesia. Whole blood was obtained via cardiac puncture. Aortic tissues and other organs were harvested separately and stored in 4% formalin for histopathological analysis or at −80 °C for biochemical experiments. All the procedures were conducted in strict accordance with the Guiding Principles for the Care and Use of Laboratory Animals and were approved by the Ethics Committee on Animal Experiments of China Agricultural University (Beijing, China) (No. AW50804202-5-3).

### 2.3. Detection of Body Composition

After 8 weeks of intervention and just prior to euthanasia, the body fat and lean mass content of the mice were analyzed using the Awake Small Animal Body Composition Analysis and Imaging System (MesoQMR-060H-I, Shanghai Numai Electronic Technology Co., Ltd., Shanghai, China). For the measurement, a 40 g animal holder was selected, and the mice were placed into the holder. The holder was then inserted into the round opening beneath the instrument, ensuring the animal was positioned correctly for the test. The non-invasive nature of the MesoQMR system allowed for real-time monitoring, providing accurate and reliable data on the body composition of the mice without the need for anesthesia, thus minimizing potential stress and confounding factors during the assessment.

### 2.4. Analysis of Serum Lipid Profiles and Inflammatory Factors

After the whole blood samples were collected from the mice, they were allowed to stand at room temperature for 2 to 3 h. Subsequently, the samples were centrifuged at 3000 rpm for 15 min to separate the serum, which was then stored at −80 °C until further analysis. The levels of total cholesterol (TC), triglycerides (TG), low-density lipoprotein cholesterol (LDL-C), and high-density lipoprotein cholesterol (HDL-C) in the serum were quantified using commercially available assay kits (S03027, S0304, S03025, and S03029; Shenzhen Raydu Life Science Co., Ltd., Shenzhen, China). The processed samples were then analyzed automatically using a biochemical analyzer (Chemray 800, Shenzhen Raydu Life Science Co., Ltd., Shenzhen, China).

### 2.5. Hematoxylin–Eosin (HE) Staining for Aorta

The aortas of the mice were carefully immersed in 4% paraformaldehyde and fixed for over 48 h. Following fixation, the tissues were embedded in paraffin and sectioned into 5 µm thick slices using a microtome. These paraffin-embedded sections were then stained with a commercially available hematoxylin and eosin (HE) staining kit (G1120, Beijing Solarbio Science and Technology Co., Ltd., Beijing, China). The staining process began with dewaxing the sections, followed by rehydration in water. The tissues were then stained with hematoxylin solution for a duration of 5–20 min, after which they were thoroughly rinsed in running tap water. To differentiate the staining, the sections were treated with differentiation solution for 3 min, followed by two washes with tap water, each lasting 2 min. The next step involved re-staining the sections with Eosin Y Aqueous Solution for 10 s to 2 min and subsequently dehydrating the samples for 2–3 min each, followed by a 1 min rinse in 100% alcohol. After dehydration, the samples were placed in a series of alcohol solutions, 75%, 85%, 95%, and 100% alcohol, and the sections were cleared using xylene and mounted with neutral gum. The stained sections were then examined under an optical microscope, and the atherosclerotic plaques were analyzed and quantified using ImageJ software (version 1.8.0, National Institutes of Health, Bethesda, MD, USA).

### 2.6. Masson’s Trichrome Stain

Weigert’s iron hematoxylin solution, fuchsin ponceau acid solution, and aniline blue solution were utilized to stain the prepared paraffin sections, following the protocol provided in the Masson’s trichrome stain kit (G1340, Beijing Solarbio Science and Technology Co., Ltd., Beijing, China). The staining process began with routine dewaxing of the paraffin sections in distilled water. Next, a 1:1 mixture of reagents A1 and A2 was applied to cover the sections, and the sections were left for 5–10 min. Afterward, the excess staining solution was washed away with distilled water, and the sections were differentiated with acid ethanol differentiation solution for 5–15 s, followed by another wash with distilled water for 30 s. The sections were then stained with Masson solution for 3–5 min, followed by a 30 s rinse in distilled water. Subsequently, a weak acid working solution was prepared by mixing distilled water and weak acid solution in a 2:1 ratio. This solution was applied to the sections for 30 s, after which the excess liquid was discarded. The sections were treated with phosphorolybdic acid solution for 1–2 min, followed by another 30 s wash with distilled water. The aniline blue stain was then applied for 1–2 min, and the sections were washed again with distilled water for 30 s. To complete the dehydration process, the sections were immersed in 95% ethanol for 2–3 s, followed by two washes in absolute ethanol for 5–10 s each. The sections were then cleared with xylene for 1–2 min, twice, and sealed with neutral gum. Finally, the stained sections were examined and photographed under an optical microscope (Leica CTR6, Leica, Wetzlar, Germany). The proportion of collagen in the stained tissue sections was analyzed and quantified using ImageJ software (version 1.8.0).

### 2.7. Analysis of MDA, IL-6, and TNF-α Levels in Serum

The levels of malondialdehyde (MDA) in serum were determined using a commercial assay kit (BC0020, Beijing Solarbio Science and Technology Co., Ltd., Beijing, China). Prior to measurement, the spectrophotometer was preheated for more than 30 min, and the distilled water was adjusted to zero. The measurement process involved preparing a measurement tube containing 600 μL of MDA working solution, 200 μL of the sample, and 200 μL of reagent 3, while a blank tube was prepared with 600 μL of MDA working solution, 200 μL of distilled water, and 200 μL of reagent 3. Both tubes were incubated in a 100 °C water bath for 60 min, followed by cooling in an ice bath. Afterward, the tubes were centrifuged at 10,000× *g* at room temperature for 10 min, and the supernatant was transferred to a 96-well plate to measure the absorbance at 532 nm and 600 nm. The differences in absorbance (ΔA_532_ and ΔA_600_) were calculated by subtracting the blank readings from the measured values, and the final value for ΔA was obtained by subtracting ΔA_600_ from ΔA_532_. For the high-fat blood blank tube, distilled water was diluted with a mixture of 33 μL distilled water and 67 μL ethanol. The MDA content (nmol/mL) was then calculated using the following formula: MDA content = 32.258 × ΔA, where ΔA represents the absorbance difference and other constants account for the total volume, sample volume, molar absorption coefficient of MDA, and dilution factor.

Additionally, for the quantitative detection of mouse interleukin-6 (IL-6) and tumor necrosis factor-α (TNF-α), an enzyme-linked immunosorbent assay (ELISA) was performed. The procedure involved coating a Corning™ Costar™ 9018 ELISA plate (Corning Incorporated-Life Sciences, Kennebunk ME, USA) with 100 µL of capture antibody in Coating Buffer and incubating the plate overnight at 4 °C. After washing the wells three times with Wash Buffer, the wells were blocked with 200 µL of ELISA/ELISPOT Diluent (1×) and incubated for 1 h at room temperature. The mouse IL-6 standard was reconstituted with distilled water, and the standard curve was prepared by performing 2-fold serial dilutions of the top standard across 8 points. Following this, 100 µL of each sample was added to the corresponding wells, and the plate was sealed and incubated for 2 h (or overnight at 4 °C for maximum sensitivity). After incubation, the detection antibody was added, followed by incubation for 1 h at room temperature. Subsequently, Avidin-HRP was added to the wells, and incubation continued for 30 min at room temperature. The plate was washed thoroughly after each step, with multiple wash cycles to ensure the effective removal of residual buffer. Finally, 100 µL of 1 × TMB Solution was added, and the plate was incubated for 15 min before adding 100 µL of Stop Solution. The absorbance was read at 450 nm, and if available, subtraction of the readings at 570 nm from those at 450 nm was performed to analyze the data.

### 2.8. Microbial Diversity Analysis

#### 2.8.1. DNA Extraction of Sample

An E.Z.N.A. Soildnakit (Omega Bio-Tek, Norcross, GA, USA) was used to extract the total genomic DNA of the microbial community from the mouse feces samples, and the specific operation was carried out according to the instructions of the kit. The extracted DNA was detected by 1% agarose gel electrophoresis, and its concentration and purity were determined by Nano Drop 2000 (Thermoscientific, Waltham, MA, USA).

#### 2.8.2. Construction of PCR Amplification and Sequencing Library

Using the extracted DNA as a template, the V3-V4 variable region of the 16SrRNA gene was amplified by PCR with primers 338 f (5′-ACTCCTACGGGAGGCCAGCAG-3′) and 806 r (5′-GGACTACHVGGGTWTCTAAT-3′) [25]. The amplification procedure was as follows: pre-denaturing at 95 °C for 3 min; 27 cycles (denaturation at 95 °C for 30 s, annealing at 55 °C for 30 s, and extension at 72 °C for 30 s); 72 °C for 10 min; and store at 4 °C (PCR instrument: T100ThermalCycler, BIO-RAD, Hercules, CA, USA). The PCR reaction system was as follows: 5× TransStartFastPfu buffer 4 μL, 2.5 mM dNTPs 2 μL, upstream primer (5 μM) 0.8 μL, downstream primer (5 μM) 0.8 μL, TransStartFastPfuDNA polymerase 0.4 μL, template DNA 10 ng, and ddH2O make up to 20 μL. The PCR products were separated by 2% agarose gel electrophoresis, purified by using a DNA gel recovery and purification kit (PCR Clean-Up Kit C01-10000, Meiji Yuhua Biomedical Technology Co., Ltd, Shanghai, China), and the recovered products were quantitatively detected by Qubit 4.0 (Thermo Fisher Scientific, Waltham, MA, USA). The sequencing library was constructed by using the NEXTFLEXRapidDNA-SeqKit, and the steps were as follows: (1) connected the linker; (2) used magnetic beads to screen and remove the self-linking fragments of the linker; (3) amplified and enriched the library template by PCR; and (4) recovered the PCR products from the magnetic beads to obtain the final library. The library was sequenced on the Illumina PE300/PE250 platform (Shanghai Meiji Biomedical Technology Co., Ltd., Shanghai, China).

#### 2.8.3. High-Throughput Sequencing Data Analysis

We used fastp [26] (v0.19.6) to control the quality of the original sequencing data: (1) The bases whose tail mass value was less than 20 were filtered, setting a 50 bp window; if the average mass value in the window was less than 20, we cut off the back-end bases and removed the reads whose length was less than 50 bp or contained N bases. (2) FLASH [27] (v1.2.11) was used to splice the pairs of PEreads into a sequence according to the overlap relationship between PEREADS, with the minimum overlap length of 10 bp and the maximum allowable mismatch ratio of 0.2, so as to screen and reject the unqualified sequences. (3) The samples were differentiated according to the barcodes and primers and the sequence direction was adjusted. The allowable mismatch number of barcodes was 0, and the maximum mismatch number of primers was two.

The amplicon sequence variant (ASV) was obtained by using DADA2 [28] plug-in in the Qiime2 process to denoise the sequence after quality control. Based on the Silva16SrRNA gene database (v138), the NaiveBayes classifier in Qiime2 was used to analyze the species taxonomy of the ASVs. At the same time, the function of the 16SrRNA gene data was predicted and analyzed by PICRUSt2 [29] (v2.2.0).

### 2.9. Statistical Analysis

The experimental results are expressed as the mean ± standard deviations (mean ± SDs). A statistical analysis was performed using GraphPad Prism software (version 8, San Diego, CA, USA). To assess the statistical differences among the three groups, one-way analysis of variance (ANOVA) followed by Tukey’s multiple comparison test was conducted. A *p*-value of less than 0.05 was considered to indicate statistical significance.

A data analysis of the intestinal flora was completed on the Meggie Bio-cloud platform, and the specific methods were as follows: mothur [30] was used to calculate the Alpha diversity index (such as Sobs and Shannon index); principal component analysis (PCA) based on the Bray–Curtis distance algorithm and non-metric multidimensional scaling analysis (NMDS) were used to evaluate the similarity of and difference in the microbial community structure among the samples. A one-way ANOVA followed by Tukey’s multiple comparison test was used to test the species abundance of the microbial communities in multiple groups of the samples, and the significance of the differences between the groups was evaluated. By LEfSe analysis [31] (LDA > 2, *p* < 0.05), the bacterial groups with significant differences between the groups from the phylum to genus level were identified. Based on the correlation coefficient of the top 10 dominant microorganisms and environmental factors, the heat map was drawn to reveal the potential relationship between the microorganisms and environmental factors.

## 3. Results

### 3.1. Determination of Feed Composition

The fat content of common peanuts and celastrol-enriched peanuts is 52.24% and 49.22%, protein content is 28.42% and 28.30%, and carbohydrate content is 16.68% and 16.26%, respectively (Table 1). The high-fat diet (H10141, Beijing HFK Bio-Technology Co., Ltd., Beijing, China) contains 21% fat, 20% protein, and 50% carbohydrates. Through calculation, the feed containing 20% regular peanuts (HFD-peanut) comprises 27.25% fat, 21.68% protein, and 43.34% carbohydrates. The feed containing 20% celastrol-enriched peanuts (HFD-Celastrol-peanut) comprises 26.64% fat, 21.66% protein, and 43.25% carbohydrates (Table 2).

### 3.2. Improvements in Body Fat and Lean Mass Percentage

In order to establish an obesity model, 6-week-old mice were fed with HFD for 10 weeks. Then, the obesity mice were divided into the HFD group, HFD-peanut group, and HFD-Cel-peanut group and then fed for an additional 8 weeks (Figure 1A). In the first two weeks of the initial modeling period and the intervention period, we detected the food intake of mice by measuring the weight of the remaining feed. The results showed that the food intake of mice in both periods tended to be stable within two weeks. This indicates that a stable intake of intervention substances has no effect on the results. In the final weight plot, we observed that the body weight of the HFD-Cel-peanut group mice was lower than the HFD group and the control peanut group (Figure 1B). These results indicated that celastrol-enriched peanut could reduce body weight.

After 18 weeks, the body fat and lean content of mice were analyzed by the Awake Small Animal Body Composition Analysis and Imaging System. The results showed that compared with the HFD-peanut group, the body fat percentage of the HFD-Cel-peanut group was significantly decreased, while the lean meat percentage was significantly increased (Figure 1C,D).

### 3.3. Celastrol-Enriched Peanut Improved the Blood Lipid Profiles

High blood lipids may cause atherosclerosis. Celastrol-enriched peanuts were shown to improve the blood lipid profiles (TG, TC, LDL-C, and HDL-C) in atherosclerotic mice. Reduced serum TG and increased HDL-cholesterol concentrations were observed in the HFD-Cel-peanut group when compared with the HFD-peanut group (Figure 2D–F). Additionally, the serum TC levels and LDL-C exhibited a reduction trend in the HFD-Cel-peanut group mice, but there is no significant difference (Figure 2E,G).

### 3.4. Celastrol-Enriched Peanut Reduces the Concentration of Oxidant-Related Markers and Inflammatory Cytokine

Malondialdehyde (MDA) is an oxidant-related marker. The celastrol-enriched peanuts reduced the MDA levels in the HFD-Cel-peanut group mice serum samples (Figure 2A), suggesting that celastrol-enriched peanut may attenuate high-fat diet-induced aortic damage. Inflammatory responses are involved in the pathophysiology of atherosclerosis. Subsequent studies confirmed the efficacy of celastrol-enriched peanut in reducing inflammation. The concentration of tumor necrosis factor-alpha (TNF-α) in the HFD-Cel-peanut group was significantly reduced compared with the HFD-peanut group (Figure 2C). Meanwhile, the serum IL-6 levels were found to be decreased in the HFD-Cel-peanut group relative to the mice in the HFD group (Figure 2B). These results suggests that the inhibition of AS by celastrol-enriched peanuts is partly due to its anti-inflammatory effect.

### 3.5. Celastrol-Enriched Peanut Reduced Atherosclerotic Plaques in the Aortic Root

A significant decrease in the atherosclerotic plaque’s region was observed by HE staining of major tissues (Figure 3A). A cross-sectional analysis of the aortic roots further demonstrated a remarkable reduction in the plaque necrotic core area in the celastrol-peanut group mice (Figure 3B,C). The data suggest that celastrol-enriched peanut as a natural antioxidant agent could alleviate atherosclerosis lesions in the aortic root.

### 3.6. Celastrol-Enriched Peanut Improved Stability in Aortic Roots

Masson staining, one of the most common methods for connective tissue staining, is used to differentiate muscle fibers (red) from collagen fibers (blue). It is generally accepted that higher collagen content in the aorta is associated with increased plaque stability and a reduced likelihood of rupture and thrombosis formation. Therefore, Masson staining results can serve as an indicator of AS plaque stability. Cel-peanut intervention displayed a marked increase in collagen fiber content (blue area; Figure 3A). A quantitative analysis further confirmed these findings, with collagen content after intervention being 54% higher than that of the HFD group and 16% higher than that of the HFD-peanut group (Figure 3D). These results suggest that HFD-Cel-peanut intervention significantly enhanced active collagen content, thereby improving plaque stability.

### 3.7. Biological Classification and Dilution Curve of Mouse Intestinal Flora

The intestinal flora of the HFD group, HFD-peanut group, and HFD-Cel-peanut group were sequenced and quality controlled. A total of 1,076,033 valid sequences were obtained, with a total base count of 450,620,837 bp and an average sequence length of 419 bp (Appendix A). For the purpose of analysis, the sequences were classified by amplicon sequence variants (ASVs), and those ASVs accounting for less than 0.001% of the total sequence number were excluded. To assess whether the sample size was sufficient to represent the majority of microorganisms in the sample, a Pan/Core analysis and dilution curve analysis (using the Shannon and Sobs indices) were performed. The results indicate that as the number of samples increases, the total number of species gradually increases, while the number of common species decreases, leading to a gradual flattening of the curves (Appendix A). Dilution curves (Appendix A) were plotted based on the Shannon and Sobs diversity indices, and all three sets of curves tended to plateau as the number of reads increased. Additionally, the Coverage (coverage rate) for each sample library was calculated, with all the values exceeding 0.999. These findings indicate that the sample size is adequate, the sequencing depth is sufficient, and the experimental requirements are met. The data effectively represent the microbial composition within the samples and are suitable for further analysis.

### 3.8. Effect of Cel-Peanut Intervention on α-Diversity of Intestinal Flora in Mice

The α-diversity of intestinal flora is commonly used to assess the richness and diversity of microbial communities. Several widely used statistical indices, including the ACE index, Sobs index, Shannon index, and Simpson index, can be employed to evaluate these characteristics. The Sobs index reflects the number of amplicon sequence variants (ASVs) that are actually observed, while the ACE index estimates the total number of ASVs present in the sample. Both the ACE (Appendix A) and Sobs (Appendix A) indices together provide an indication of community richness. The results show no significant differences between the two indices after cel-peanut intervention, suggesting that this intervention does not affect the richness of the intestinal flora. The Shannon index (Appendix A) and Simpson index (Appendix A) are commonly used to represent regional biodiversity in ecology and to estimate microbial diversity within samples. Generally, higher Shannon values and lower Simpson values indicate greater α-diversity. The results show that the Shannon index did not change significantly following cel-peanut intervention, while the Simpson index increased significantly. These findings suggest that while the cel-peanut intervention did not alter community richness, it led to an enrichment of the dominant bacterial communities.

### 3.9. Effect of Cel-Peanut Intervention on β-Diversity

Principal component analysis (PCA) was performed to evaluate the community composition of the samples, providing insight into the differences between them. In the PCA plot (Figure 4A), the closer the distance between two samples, the more similar their species composition. The results indicated that samples within the same group were relatively close to each other, clustering together in the same region of the plot. In contrast, the distances between the HFD, HFD-peanut, and HFD-Cel-peanut groups were noticeably greater, with distinct separation along the PC1 axis. To further explore community differences, a non-metric multidimensional scaling (NMDS) analysis was conducted, using a two-dimensional scatter plot to visualize variations in the species composition between the samples or groups. The Bray–Curtis algorithm was employed to quantify the degree of aggregation or dispersion of the sample communities, based on the distances between them. Combined with the stress value (stress < 0.2, indicating a meaningful plot) and statistical analysis using the ANOSIM test, the results demonstrated significant differences in the community composition between the control and treatment groups. The closer two sample points were, the more similar their species composition. The findings revealed that several samples within the same group were evenly distributed along the PC1 axis, with a stress value of 0.107 and a *p*-value of less than 0.05, indicating a significant difference between the groups on the PC2 axis (Appendix A).

### 3.10. Effect of Cel-Peanut Intervention on the Composition of Intestinal Flora

The results above indicate that cel-peanut intervention affects the diversity of intestinal flora. As shown in the Venn diagram (Figure 4B), there are 101 common species shared among the HFD group, HFD-peanut group, and HFD-Cel-peanut group, with 8 species unique to the HFD group, 2 species unique to the HFD-peanut group, and 5 species unique to the HFD-Cel-peanut group.

Based on these findings, a further analysis of the flora composition of the three groups of mice was conducted. The composition of the intestinal flora was examined at the phylum level (Figure 4C). The dominant phyla across all three groups included *Bacillota*, *Bacteroidota*, *Thermodesulfobacteriota*, *Actinomycetota*, and *Patescibacteria*. A detailed comparison of the proportions of different bacterial phyla revealed that cel-peanut intervention primarily impacted *Bacillota*, *Bacteroidota*, and *Thermodesulfobacteriota*. Compared with the HFD group, *Bacillota* decreased from 63.33% to 45%, while *Bacteroidota* increased from 29.71% to 51.51%. When compared with the HFD-peanut group, *Bacillota* decreased from 53.42% to 45%, *Bacteroidota* increased from 25.66% to 51.51%, and *Thermodesulfobacteriota* decreased from 12.49% to 1.19%. The proportions of other bacterial phyla were relatively small, showing minimal differences from the HFD group after intervention. Additionally, compared with the HFD-peanut group, *Actinomycetota* and *Patescibacteria* decreased by approximately 1.5% and 3.5%, respectively.

The composition of the intestinal microbiota at the genus level was analyzed (Figure 4D). Compared with the HFD group, species that showed a significant decrease (greater than 1%) after cel-peanut intervention primarily included the following: *Allobaculum*, which decreased from 15.91% to 4.41%; *Dubosiella*, which decreased from 5.34% to 0%; and *Ileibacterium*, which decreased from 5.7% to 0%. Conversely, species that increased significantly (greater than 1%) included the following: *norank_f__Muribaculaceae*, which increased from 19.52% to 29.05%; *Bacteroides*, which increased from 3.41% to 9.53%; and *Ligilactobacillus*, which increased from 2.76% to 6.98%.

Compared with the HFD-peanut group, species showing a significant decrease (greater than 1%) after cel-peanut intervention primarily included the following: *Lachnospiraceae*, which decreased from 7.97% to 4.16%; *Desulfovibrio*, which decreased from 10.96% to 0.8%; *Dubosiella*, which decreased from 4.16% to 0%; *Candidatus_Saccharimonas*, which decreased from 4.35% to 0.9%; and *Clostridium*, which decreased from 3.1% to 0.6%. Species that increased significantly (greater than 1%) included the following: *norank_f__Muribaculaceae*, which increased from 16.13% to 29.05%; *Bacteroides*, which increased from 4.53% to 9.53%; *Ligilactobacillus*, which increased from 5.02% to 6.98%; and *Alistipes*, which increased from 1.3% to 2.6%.

### 3.11. Effect of Cel-Peanut Intervention on Characteristic Bacterial Genera

The LEfSe multi-level species difference analysis was performed at the genus level for the HFD group, HFD-peanut group, and HFD-Cel-peanut group. The Linear Discriminant Analysis (LDA) score was used to assess the impact of the species on the differences observed and to identify characteristic bacterial genera. The results (Figure 4E) revealed that the HFD group was significantly enriched in *g_Allobaculum*, *g_Dubosiella*, *g_Enterococcus*, and *g_norank_o_RF39*, among others. In contrast, *Desulfovibrio*, *g_Mucispirillum*, *g_Lachnospiraceae_UCG-006*, *g_Lachnospiraceae_NK4A136_group*, *g_Coriobacteriaceae_UCG-002*, and *g_Muribaculum* were significantly accumulated in the HFD-peanut group. The HFD-Cel-peanut group showed a significant enrichment in *g_Bilophila* and *g_norank_o_Clostridia_vadinBB60_group*.

### 3.12. Effect of Cel-Peanut Intervention on the Difference of Intestinal Flora

To evaluate the differences in the intestinal flora caused by cel-peanut, we conducted one-way ANOVA followed by Tukey’s multiple comparison test on the intestinal flora of the three groups at both the phylum and genus levels to analyze significant differences in species abundance among the groups (Figure 4F).

At the phylum level, the HFD-peanut group was significantly enriched in *Thermodesulfobacteriota* and *Deferribacterota* (*p* < 0.05). However, after intervention with HFD-Cel-peanut, the relative abundance of these two phyla decreased significantly (*p* = 0.004; *p* = 0.02), even falling below the levels observed in the HFD group. At the genus level, the HFD-peanut group was significantly enriched in *Desulfovibrio*, *Lachnospiraceae_NK4A136_group*, and, to a lesser extent, *Mucispirillum* and *g_norank_o_RF39*. Following the HFD-Cel-peanut intervention, the relative abundance of *Desulfovibrio* and *Mucispirillum* decreased significantly (*p* = 0.004; *p* = 0.02), while the abundance of *Lachnospiraceae_NK4A136_group* remained comparable with that in the HFD-peanut group.

These results indicate that cel-peanut intervention can significantly reduce the relative abundance of harmful bacteria while maintaining or increasing the abundance of beneficial bacteria, thus improving the composition of the intestinal flora.

### 3.13. Correlation Analysis Between Intestinal Flora and Clinical Factors

#### 3.13.1. Correlation Between Intestinal Flora and Blood Lipids

Changes in intestinal flora are strongly correlated with serum triglycerides (TGs), total cholesterol (TC), low-density lipoprotein cholesterol (LDL-C), and high-density lipoprotein cholesterol (HDL-C) concentrations (Figure 5A). TG levels are positively correlated with *Desulfovibrio*, *Candidatus_Saccharimonas*, *Clostridium*, and others while being negatively correlated with *Rikenellaceae_RC9_gut_group*, *Clostridium*, *Butyricimonas*, and others. TC levels exhibit a positive correlation with *Candidatus_Saccharimonas*, *Desulfovibrio*, *Dubosiella*, and others and a negative correlation with *norank_f_Prevotellaceae*, *Rikenellaceae_RC9_gut_group*, and others. LDL-C levels are positively correlated with *Desulfovibrio*, *Clostridium*, and others and negatively correlated with *norank_f_Prevotellaceae*, *Butyricimonas*, and others. HDL-C levels are positively correlated with *Rikenellaceae_RC9_gut_group*, *norank_f_Prevotellaceae*, *Butyricimonas*, and others while being negatively correlated with *Desulfovibrio*, *Candidatus_Saccharimonas*, *Clostridium*, and others.

#### 3.13.2. Correlation Between Intestinal Flora and Oxidative Factors, Inflammatory Factors

The correlation between the oxidative factors; inflammatory factors such as MDA, IL-6, and TNF-α; and the intestinal flora was investigated (Figure 5B). Among the antioxidant indicators, MDA was positively correlated with *Desulfovibrio*, *Candidatus_Saccharimonas*, and *Clostridium*, while it was negatively correlated with *norank_f_Prevotellaceae* and *Rikenellaceae_RC9_gut_group*. The inflammatory factor IL-6 was positively correlated with *Allobaculum*, *Alistipes*, and *Ileibacterium*, whereas TNF-α was strongly positively correlated with *Lachnospiraceae_UCG_006* and *Turicibacter*.

#### 3.13.3. Correlation Between Intestinal Flora and AS Lesion Parameters

The correlation analysis of the intestinal flora and AS lesion parameters (including AS plaque area and collagen content) revealed (Figure 5C) that the plaque area was significantly positively correlated with *Turicibacter* and *Ileibacterium* and negatively correlated with *Rikenellaceae_RC9_gut_group*, *norank_f__Muribaculaceae*, and *Butyricimonas*. In contrast, thecollagen content was significantly positively correlated with *Lachnospiraceae_UCG-006*, *Rikenellaceae_RC9_gut_group*, *Ligilactobacillus*, and others, while it was negatively correlated with *Dubosiella*, *Ileibacterium*, and *Allobaculum*. Since collagen content is a key indicator of plaque stability, these findings suggest that plaque stability is closely related to alterations in the bacterial flora. Overall, the correlations reveal an inverse relationship between plaque area and collagen content in relation to bacterial flora composition. This inverse trend may be attributed to the progression of lesions into mid-to-late stages, during which plaque stability diminishes and collagen content declines, resulting in contrasting patterns between plaque area and bacterial composition.

### 3.14. Prediction of Intestinal Flora Function

To explore the significant role of intestinal flora in the HFD-Cel-peanut group, the KEGG pathway of PICRUSt (Phylogenetic Investigation of Communities by Reconstruction of Unobserved States) was used to predict the potential functions of the bacteria and to examine the correlation between differential flora and predicted functional pathways.

The results are presented for pathway levels 1, 2, and 3. At pathway levels 1 and 2, the most abundant pathways were related to metabolism and global overview maps, which were considerably more abundant than other pathways (Figure 5D,E). At pathway level 3, the metabolic pathway exhibited the highest abundance, followed by the biosynthesis of secondary metabolites (Figure 5F). These findings suggest that cel-peanut intervention holds promising potential for applications in metabolic pathways.

## 4. Discussion

With the popularity of the anti-inflammatory diet, the role of natural products in preventing and treating metabolic disorders has gained significant attention [32]. Celastrol, a triterpenoid extracted from the *Tripterygium wilfordii*, has emerged as a potent agent due to its anti-inflammatory and anti-obesity properties. Prior studies indicate that celastrol significantly reduces inflammation by inhibiting pro-inflammatory cytokines, such as IL-6, while also mitigating the harmful effects of oxidized low-density lipoprotein (oxLDL), both of which are crucial in the pathogenesis of atherosclerosis [33]. Furthermore, celastrol has been demonstrated to promote beneficial vascular remodeling, a process that is vital for maintaining cardiovascular health and mitigating diseases such as atherosclerosis [34]. Despite its promising benefits, several challenges hinder its clinical application. On one hand, celastrol is lipophilic, which limits its solubility in water and subsequently reduces its bioavailability. This poses restrictions for formulating effective delivery systems that maximize its absorption and utilization in the body. On the other hand, as an alkaloid, it still has some pharmacokinetic limitations and undesirable side effects that need to be overcome. The liver, kidney, cholangiocytes, heart, ear, and reproductive system may be affected by these toxic effects [35]. Recent evidence indicated that celastrol may be strictly limited due to the occurrence of severe side effects, which could be reduced by structural modification [36]. Delivery systems such as nanotechnology have been tried to diminish the potential toxicity of celastrol. An injectable thermosensitive micelle–hydrogel hybrid system loaded with celastrol was reported to be able to sustain and prolong the release of celastrol to inhibit posterior capsule opacification and had no apparent tissue toxicity [36,37]. However, in the context of food and pharmaceuticals, one of the significant challenges in utilizing bioactive compounds in nanoparticle form is the limited understanding of their cytotoxicity thresholds [38].

To tackle these problems, we designed a novel approach involving the selection and cultivation of celastrol-enriched peanuts. This method consists of distant hybridization and metabolomics analysis to enhance the natural content of celastrol in peanuts, offering a more effective delivery system while minimizing potential side effects. Meanwhile, it is essential to find out which combination of diet and medication is the most favorable and appropriate for each potential chronic disease [39]. To achieve a balance in our animal feed formulation, we monitored some key parameters, including the following: (1) to achieve the effect of making animals obese and ensure the effectiveness of their nutritional intervention; (2) to ensure that the macronutrient content of the new feed does not deviate too much from the original composition of the high-fat feed. Finally, replacing 20% of the high-fat feed with peanuts is an acceptable option.

Numerous studies have demonstrated that atherosclerosis is a chronic inflammatory disease, and high lipid levels can accelerate the development of atherosclerosis. These results suggested that celastrol-enriched peanut could affect the body fat and lean mass percentage and play a role in reducing fat and improving lean mass percentage. Our findings also clearly indicate that the inclusion of celastrol-enriched peanuts can significantly improve relevant biomarkers associated with this condition. Firstly, celastrol-enriched peanut improved the blood lipid levels of HFD-induced *ApoE*^−/−^ mice. Specifically, it led to a significant decrease in the serum TG concentration and an increase in HDL-C. Since lipid dysregulation plays a pivotal role in the progression of atherosclerotic plaques [3], the improvement in atherosclerosis after the intervention with celastrol-enriched peanut may be partly attributed to its beneficial effect on blood lipids. Secondly, the intervention of celastrol-enriched peanut decreased the expression of oxidation-related markers and inflammatory factors in mice. Celastrol-enriched peanut reduced the levels of serum MDA, IL-6, and TNF-*α*. A convincing body of experimental and clinical data indicates that inflammation participates fundamentally in atherogenesis [40]. Our findings suggest that celastrol-enriched peanut can improve lipid peroxidation and prevent cell damage caused by inflammation.

As mentioned above, recent studies have highlighted the significant role of intestinal flora in the development of atherosclerotic diseases. Following cel-peanut intervention, the composition of the intestinal flora in mice changed significantly. At the phylum level, AS mice induced by a high-fat diet (HFD) were predominantly composed of *Bacillota* and *Bacteroidota*, whereas cel-peanut intervention significantly reduced the relative abundance of *Bacillota* and increased the relative abundance of *Bacteroidota*. Previous studies have indicated that downregulation of *Bacillota* and upregulation of *Bacteroidota* can help regulate lipid metabolism disorders [41]. At the genus level, Zheng et al. [42] observed a significant increase in the relative abundance of *Allobaculum* and the expression of ANGPTL4 in HFD mice. The increased ANGPTL4 expression inhibits lipid absorption. After cel-peanut intervention, the relative abundance of *Allobaculum* decreased, suggesting that cel-peanut may improve AS by promoting lipid absorption. Additionally, *Desulfovibrio* can absorb sulfate and produce hydrogen sulfide (H_2_S), which is toxic to intestinal epithelial cells and may contribute to gastrointestinal diseases [43]; *Mucispirillum* is associated with intestinal inflammation, and its abundance can improve colitis symptoms through a fibrotic diet [44]. In the HFD-Cel-peanut group, the relative abundance of *norank_f_Muribaculaceae*, *Bacteroides*, *Ligilactobacillus*, and *Lachnospiraceae* was significantly increased. Previous studies have shown that *norank_f_Muribaculaceae* can alleviate intestinal inflammation [45]. *Bacteroides* plays a key role in intestinal metabolism, including the utilization of nitrogen-containing substances, fermentation of carbohydrates, and biotransformation of bile acids and other sterols. These changes may aid the digestion of food and nutrient production and support the growth of other bacteria to maintain intestinal homeostasis [46]. A high-fat diet reduces the relative abundance of *Ligilactobacillus*, but cel-peanut intervention significantly increases its abundance, thereby restoring the intestinal flora structure [47]. Furthermore, *Lachnospiraceae* is one of the primary bacteria that ferments dietary fiber to produce short-chain fatty acids (SCFAs) like butyrate, which serve as an important energy source for colon cells. *Butyrate* also has an anti-inflammatory property, which improves intestinal barrier integrity and promotes overall intestinal health [48].

To further investigate the relationship between intestinal flora and AS-related parameters, such as blood lipids, oxidative stress markers, inflammatory factors, lesion area, and collagen content, we analyzed the correlation between the abundance of intestinal flora and these clinical indicators. The results showed that after cel-peanut intervention, the relative abundance of *Desulfovibrio* and *Isobacteria* decreased significantly. Collagen content was negatively correlated with *Dubosiella* and *Ileibacterium* (*p* < 0.05), while the plaque area was positively correlated with *Rikenellaceae_RC9_gut_group*, *norank_f__Muribaculaceae,* and *Butyricimonas* (*p* < 0.05), suggesting a decrease in plaque area and an increase in collagen content.

## 5. Conclusions

Collectively, cel-peanut may increase the abundance of beneficial bacteria, decrease the abundance of harmful bacteria, regulate lipid metabolism, reduce inflammation, and improve the function of intestinal microflora, thereby promoting intestinal health, maintaining homeostasis, and ultimately slowing the progression of AS. These findings suggest that celastrol-enriched peanuts could serve as a beneficial component of an anti-inflammatory diet, potentially aiding in weight reduction and alleviating atherosclerosis. However, the specific mechanisms through which how celastrol influences atherosclerosis remain unclear, underscoring the need for further investigation in this area. Additionally, more clinical trials are necessary to assess the long-term effects and safety of celastrol in humans, particularly among diverse populations with varying levels of metabolic disorders.

In future research, we hope to utilize gradient dosing and varied durations of intervention to better determine the optimal dose and duration of action for celastrol. We will also compare the effects of celastrol-enriched peanuts with those of celastrol alone to highlight the advantages and disadvantages of the peanut carrier strategy. Moreover, we aim to explore the mechanisms by which celastrol and other bioactive compounds in peanuts work synergistically to achieve therapeutic effects. It may provide more reference for other researchers.

## Figures and Tables

**Figure 1 nutrients-17-01418-f001:**
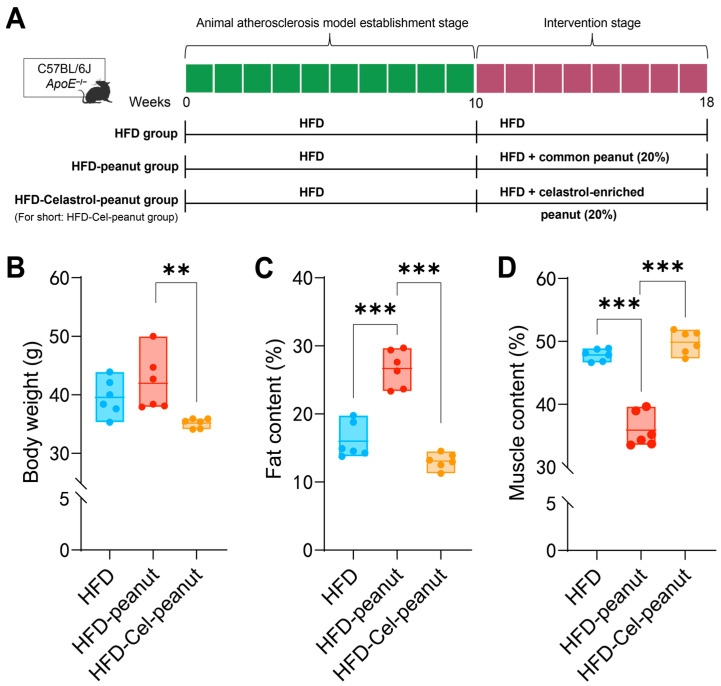
(**A**) Scheme of celastrol-enriched peanut intervention in HFD-induced atherosclerosis in *ApoE*^−/−^ mice. (**B**) Effects of celastrol-enriched peanut on body weight in HFD-induced *ApoE*^−/−^ mice. Body weight in HFD group, HFD-peanut group, and HFD-Cel-peanut group. Effects of celastrol-enriched peanuts on body composition in HFD-induced *ApoE*^−/−^ mice. Fat content (**C**) and muscle content (**D**) percentage in HFD group, HFD-peanut group, and HFD-Cel-peanut group. Results are presented as means ± SDs, and n = 6 in each group. One-way ANOVA with Tukey’s multiple comparison test; ** *p* < 0.01 and *** *p* < 0.001.

**Figure 2 nutrients-17-01418-f002:**
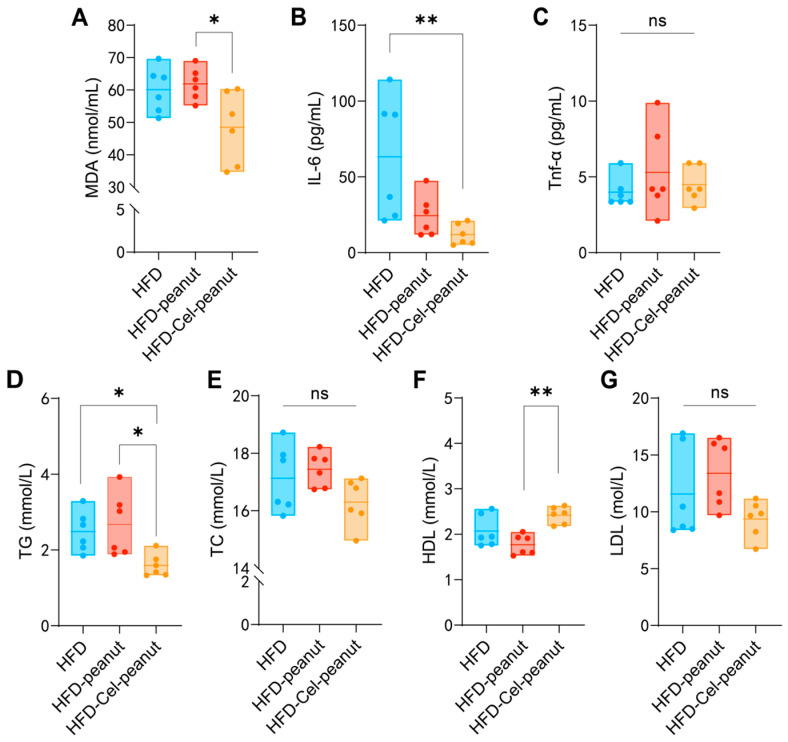
Effect of celastrol-enriched peanut on serum lipids in HFD-induced *ApoE*^−/−^ mice. TG (**A**), TC (**B**), HDL-C (**C**), and LDL-C (**D**) levels of serum in dandelion polysaccharides and saline group. Effect of celastrol-enriched peanuts on antioxidant markers and inflammatory cytokines in HFD-induced *ApoE*^−/−^ mice. MDA, TNF-α, and IL-6 concentrations (**E**–**G**) in serum. Results are presented as means ± SDs, and n = 6 in each group. One-way ANOVA with Tukey’s multiple comparison test; * *p* < 0.05 and ** *p* < 0.01.

**Figure 3 nutrients-17-01418-f003:**
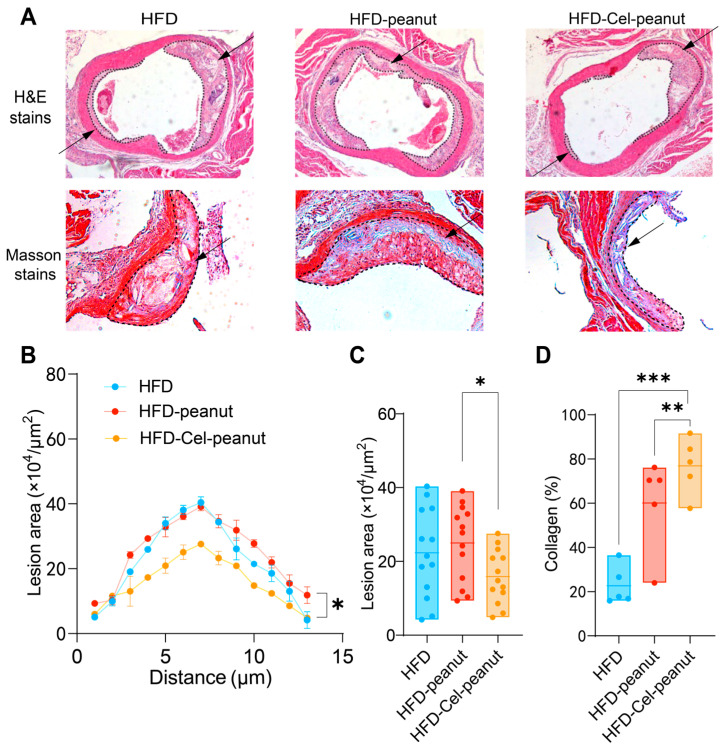
Effect of celastrol-enriched peanut on atherosclerotic lesion in HFD-induced *ApoE*^−/−^ mice. Representative images for HE staining and Masson staining (**A**), curve chart (**B**), and quantitative chart of atherosclerotic lesion area by HE staining (**C**) and collagen area by Masson staining (**D**). For HE staining, results are presented as means ± SDs (n = 13 in each group). The arrows in the figure indicate the plaque area formed in the aorta. Each point represents the average plaque area of four mice, with two slices counted per mouse. For Masson staining, results are presented as means ± SDs (n = 5 in each group). * *p* < 0.05, ** *p* < 0.01, and *** *p* < 0.001; one-way ANOVA with Tukey’s multiple comparison test.

**Figure 4 nutrients-17-01418-f004:**
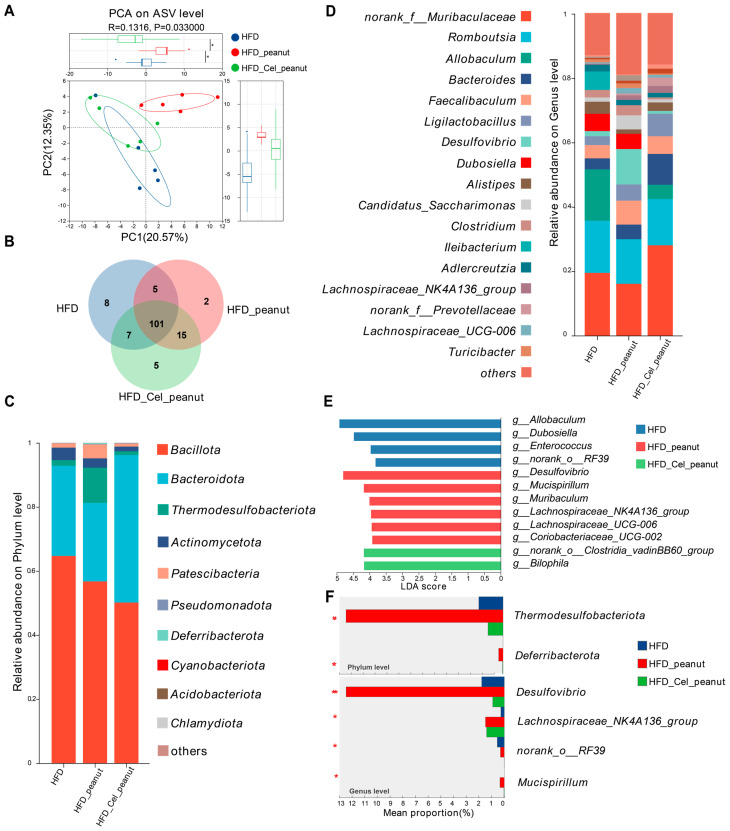
Effect of cel-peanut intervention on β-diversity of intestinal flora in *ApoE*^−/−^ mice. (**A**) PCA analysis of intestinal microbiota and box plot based on PC1 and PC2 axis to evaluate sample community composition. (**B**) Intestinal flora species number Venn chart. (**C**) Intestinal bacterial community composition (phylum level). (**D**) Effect of cel-peanut intervention on intestinal flora composition (genus level). The horizontal coordinate represents grouping, and the vertical coordinate represents the relative abundance of the bacterial population (genus level). (**E**) Effect of cel-peanut intervention on characteristic bacterial genera in the intestinal flora of AS mice. (**F**) Effect of cel-peanut intervention on multi-group comparative analysis of species differences at the phylum level and genus level. In the figure, n = 5 means there are 5 mice; * *p* < 0.05 and ** *p* < 0.01, with significant differences.

**Figure 5 nutrients-17-01418-f005:**
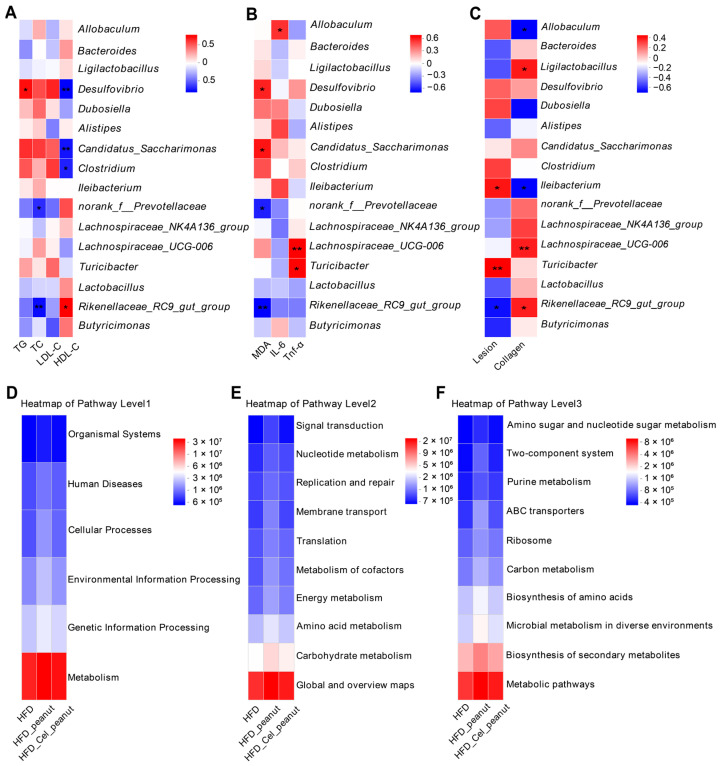
(**A**) Correlation analysis between intestinal flora and blood lipids. (**B**) Correlation analysis between intestinal flora and oxidative and inflammatory factors. (**C**) Correlation analysis between lesion area and collagen percentage. KEGG analysis of intestinal flora at pathway levels 1, 2, and 3. (**D**) KEGG results of intestinal flora at pathway level 1, (**E**) KEGG results of intestinal flora at pathway level 2, and (**F**) KEGG results of intestinal flora at pathway level 3. Red indicates positive correlation, blue indicates negative correlation, and the darker the color, the stronger the correlation, and the lighter the correlation is, the weaker the correlation. In the figure, * *p* < 0.05, and ** *p* < 0.01, with significant differences.

**Table 1 nutrients-17-01418-t001:** Nutrient composition of common peanut and celastrol-enriched peanut.

	Common Peanut	Celastrol-Enriched Peanut
Fat	52.24%	49.22%
Protein	28.42%	28.30%
Carbohydrate	16.68%	16.26%
Celastrol content	0.57 μg/kg	1029.21 μg/kg

**Table 2 nutrients-17-01418-t002:** Nutrient composition of three feeds.

	HFD Group	HFD-Peanut Group	HFD-Cel-Peanut Group
Fat	21.00%	27.25%	26.64%
Protein	20.00%	21.68%	21.66%
Carbohydrate	50.00%	43.34%	43.25%

## Data Availability

The original contributions presented in this study are included in the article/Appendix A; further inquiries can be directed to the corresponding authors.

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
