# Peer review of "Effects of Celastrol-Enriched Peanuts on Metabolic Health and the Development of Atherosclerosis"

_nutrients, 2025, doi:10.3390/nu17091418_

Round 1
Reviewer 1 Report
Comments and Suggestions for Authors
Dear Author,
Appreciation is extended for the submission of your research.
This investigation devised an innovative peanut variety (cel-peanut) fortified with celastrol, a substance possessing anti-inflammatory and antioxidant capabilities, aimed at combating atherosclerosis (AS). Cel-peanut enhanced celastrol's absorption and diminished its harmful effects. Utilizing a murine model, the cel-peanut formulation notably ameliorated lipid profiles, lessened atherosclerotic plaque formation, and augmented collagen deposition within plaques. Furthermore, it mitigated inflammatory responses and influenced the gut microbiome. The results indicate cel-peanut's promising therapeutic capacity against AS via diverse pathways, presenting a novel dietary approach.
The study is meticulously organized and clearly demonstrates a substantial undertaking by the researchers. The subject matter holds relevance for a broad audience, and the clinical implications of this work are widely recognized, especially given the advantageous dietary effects.
Regards
Comments on the Quality of English LanguageDear Editor, I find the study innovative for its proposal of a peanut cultivar enriched with celastrol through hybridization. This approach exploits the beneficial properties of a natural compound, improving its bioavailability and reducing its toxicity. Furthermore, the study is multidisciplinary as it integrates approaches from plant biology, metabolomics, nutrition, and pharmacology, demonstrating a holistic view of research. This study paves the way for new nutritional intervention strategies for the prevention and treatment of atherosclerosis.
Overall, this study represents a significant contribution to atherosclerosis research. The innovative approach and promising results open new perspectives for the prevention and treatment of this disease.
Regards
Author Response
Comments 1:
This investigation devised an innovative peanut variety (cel-peanut) fortified with celastrol, a substance possessing anti-inflammatory and antioxidant capabilities, aimed at combating atherosclerosis (AS). Cel-peanut enhanced celastrol's absorption and diminished its harmful effects. Utilizing a murine model, the cel-peanut formulation notably ameliorated lipid profiles, lessened atherosclerotic plaque formation, and augmented collagen deposition within plaques. Furthermore, it mitigated inflammatory responses and influenced the gut microbiome. The results indicate cel-peanut's promising therapeutic capacity against AS via diverse pathways, presenting a novel dietary approach.
The study is meticulously organized and clearly demonstrates a substantial undertaking by the researchers. The subject matter holds relevance for a broad audience, and the clinical implications of this work are widely recognized, especially given the advantageous dietary effects.
Response 1: We sincerely thank you for such positive and encouraging comment.
Point 1:
Dear Editor, I find the study innovative for its proposal of a peanut cultivar enriched with celastrol through hybridization. This approach exploits the beneficial properties of a natural compound, improving its bioavailability and reducing its toxicity. Furthermore, the study is multidisciplinary as it integrates approaches from plant biology, metabolomics, nutrition, and pharmacology, demonstrating a holistic view of research. This study paves the way for new nutritional intervention strategies for the prevention and treatment of atherosclerosis. Overall, this study represents a significant contribution to atherosclerosis research. The innovative approach and promising results open new perspectives for the prevention and treatment of this disease.
Response 1: We sincerely thank you for such positive and encouraging comment.
Reviewer 2 Report
Comments and Suggestions for Authors
Effects of Celastrol-Enriched Peanuts on Metabolic Health and the Development of Atherosclerosis
The authors used a high-fat diet-induced ApoE−/− atherosclerotic mouse model to systematically evaluate the anti-atherosclerotic (AS) effects and mechanisms of celastrol-enriched peanuts (cel-peanut).
Major changes are necessary.
Comments and Suggestions:
Introduction: The authors should briefly explain the importance of inflammation in atherosclerosis rather than simply stating that it is important.
Line 55: Please avoid using the phrase "diet to modulate inflammation." The recommendation is not solely for inflammation but rather for an overall metabolic impact. A more precise statement would be:
“Low consumption of salt and foods of animal origin, along with increased intake of plant-based foods—whole grains, fruits, vegetables, legumes, and nuts—are linked to a reduced risk of atherosclerosis. Similarly, replacing butter and other animal/tropical fats with olive oil and other unsaturated-fat-rich oils contributes to cardiovascular health.”
This type of diet influences the entire metabolism, not just inflammation.
Line 65: The sentence “It even treats several cancers” is misleading. Based on reference 14, celastrol is not yet approved for cancer treatment. Instead, it has a theoretical basis for clinical development and application. Please revise this sentence to avoid implying that celastrol is an approved cancer drug.
Methods Section:
Administration Details: Please specify how celastrol was administered to the animals and at what dose.
Extraction Method: The authors should describe the extraction methods used for obtaining celastrol.
Nutritional Information: While the nutritional composition of lipids and protein in peanuts and celastrol-enriched peanuts is provided, what about carbohydrates? This information should be included.
Results and Discussion:
Contradictory Statement: The authors state in the introduction that "Peanut is a favorable food for reducing cardiovascular risk factors." However, Figure 1B suggests the opposite—showing increased body weight, fat mass, and decreased lean mass. Could you clarify this discrepancy?
Inflammatory Markers: The same concern applies to Il-6 and Tnf-α, as the data seem to indicate that they induce inflammation rather than reduce it. Please provide an explanation.
Control Group: Why is there no control group without obesity? A non-obese control would help clarify the independent effects of celastrol.
It the abstract the authors write “The findings indicate that cel-peanut significantly reduced serum levels of triglycerides (TG), total cholesterol (TC), and low-density lipoprotein cholesterol (LDL-C), while increasing high-density lipoprotein cholesterol (HDL-C).” however, this is no significant changes in LDL, HDL.
Author Response
Comments 1: Introduction: The authors should briefly explain the importance of inflammation in atherosclerosis rather than simply stating that it is important.
Response 1: Thanks for this suggestion.I will briefly explain the importance of inflammation in atherosclerosis and add it to the introduction
Atherosclerosis (AS) is a chronic inflammatory process triggered by endothelial injury. Its pathogenesis involves: minor endothelial damage increasing permeability and initiating inflammation, allowing low-density lipoprotein infiltration and oxidative modification in the arterial wall; monocyte recruitment and transformation into macrophages and foam cells; platelet adhesion and growth factor release; migration and proliferation of medial smooth muscle cells into the intima, synthesizing extracellular matrix (collagen and proteoglycans); and progressive lipid deposition in macrophages and smooth muscle cells. This chronic and complex process involves a continuous vicious cycle of lipid accumulation, inflammatory responses, and cellular proliferation, ultimately forming lipid-rich cores and fibrotic plaques.(Line 47-55).
Reference: Moss J W, Ramji D P. Nutraceutical therapies for atherosclerosis[J]. Nat Rev Cardiol, 2016,13(9):513-532.
Comments 2: Line 55: Please avoid using the phrase "diet to modulate inflammation." The recommendation is not solely for inflammation but rather for an overall metabolic impact. A more precise statement would be:“Low consumption of salt and foods of animal origin, along with increased intake of plant-based foods—whole grains, fruits, vegetables, legumes, and nuts—are linked to a reduced risk of atherosclerosis. Similarly, replacing butter and other animal/tropical fats with olive oil and other unsaturated-fat-rich oils contributes to cardiovascular health.”This type of diet influences the entire metabolism, not just inflammation.
Response 2: Thanks for this suggestion.I will use more accurate language to describe.
An increase in the intake of plant-based foods (whole grains, fruits, vegetables, legumes, and nuts) is associated with a reduced risk of atherosclerosis. Similarly, substituting butter and other animal/tropical fats with oils rich in unsaturated fats can benefit cardiovascular health. This type of diet affects the entire metabolism. (Line 62-65)
Comments 3: Line 65: The sentence“It even treats several cancers” is misleading. Based on reference 14, celastrol is not yet approved for cancer treatment. Instead, it has a theoretical basis for clinical development and application. Please revise this sentence to avoid implying that celastrol is an approved cancer drug.
Response 3: Thanks for this suggestion.I apologize for my inaccurate statement, I will delete that sentence. (Line 72)
Comments 4: Administration Details: Please specify how celastrol was administered to the animals and at what dose.
Response 4: Thanks for this suggestion.The dosage is specified on line 132-135, but it is still not clear enough. I will add more detailed instructions:
Three types of animal feed were used in this experiment. 1) High-fat purified diet (H10141, purchased from Beijing HFK Bio-Technology Co., Ltd) with a fat energy ratio of 41%, supplemented with 1.5% cholesterol, was used to induce high-fat and atherosclerosis models in mice. 2) H10141 mixed with 20% regular peanuts per kilogram. 3) H10141 mixed with 20% celastrol-enriched peanuts per kilogram. The common peanuts (Luhua 11, celastrol content: 0.57 μg/kg) and celastrol-enriched peanuts (celastrol content: 1029.21 μg/kg) used in this study were provided by Hainan Misheng Biotechnology Co., Ltd. (Hainan, China). (Line 110-117)
All experimental animals were fed a high-fat diet (HFD) for 10 weeks to induce obesity and atherosclerosis. Mice were randomly divided into three groups (6 mice per group): the blank high-fat diet group (HFD group), the high-fat diet with regular peanut control group (HFD-peanut group), and the high-fat diet with celastrol-enriched peanut intervention group (HFD-Cel-peanut group). Each group was fed their corresponding diets accordingly. (Line 123-128)
Comments 5: Extraction Method: The authors should describe the extraction methods used for obtaining celastrol.
Response 5: In this experiment, whole peanuts were added to the feed without any extraction process.
Comments 6: Nutritional Information: While the nutritional composition of lipids and protein in peanuts and celastrol-enriched peanuts is provided, what about carbohydrates? This information should be included.
Response 6: Thanks for this suggestion. I will supplement the carbohydrate information, the complete article is as follows:
The fat content of common peanuts and celastrol-enriched peanuts is 52.24% and 49.22% respectively, the protein content is 28.42% and 28.30% respectively, and the carbohydrate content is 16.68% and 16.26% respectively. (Line 299-308)
Comments 7: Contradictory Statement: The authors state in the introduction that "Peanut is a favorable food for reducing cardiovascular risk factors." However, Figure 1B suggests the opposite—showing increased body weight, fat mass, and decreased lean mass. Could you clarify this discrepancy?
Response 7: Thanks for this suggestion. Peanuts are rich in unsaturated fatty acids, which are beneficial for reducing the risk of cardiovascular diseases. However, they are still high-fat foods, and excessive consumption can lead to obesity. Therefore, my statement in the introduction that "Peanut is a favorable food for reducing cardiovascular risk factors" is one-sided, and I will delete this sentence.
Comments 8: Inflammatory Markers: The same concern applies to Il-6 and Tnf-α, as the data seem to indicate that they induce inflammation rather than reduce it. Please provide an explanation.
Response 8: Thanks for this suggestion.
Generally speaking, peanuts are rich in oleic acid (Omega-9), which is the same as the main fatty acid in olive oil, and has anti-inflammatory properties that may reduce the risk of cardiovascular diseases. They contain antioxidants such as vitamin E and polyphenols (like resveratrol), which can neutralize free radicals and reduce oxidative stress and chronic inflammation. Peanuts are also high in fiber, which can promote gut health and indirectly inhibit inflammation by regulating the balance of gut microbiota.However, peanuts contain a high amount of Omega-6 (linoleic acid), while modern diets are generally deficient in Omega-3. Therefore, excessive intake of Omega-6 may lead to an imbalance in the Omega-6/Omega-3 ratio, promoting the release of pro-inflammatory factors (such as IL-6, TNF-α).
Reference: Kang, J., Wang, J., Wu, L.et al. Fat-1 mice convert n-6 to n-3 fatty acids. Nature 427, 504 (2004). https://doi.org/10.1038/427504a.
Some studies also suggest that peanut lectin may increase intestinal permeability, allowing bacterial toxins to enter the bloodstream and trigger chronic inflammation.
Reference: Peanut ingestion increases rectal proliferation in individuals with mucosal expression of peanut lectin receptor Ryder, Stephen D. et al. Gastroenterology, Volume 114, Issue 1, 44-49.
Comments 9: Control Group: Why is there no control group without obesity? A non-obese control would help clarify the independent effects of celastrol.
Response 9: Thanks for this suggestion. This is the limitation of the present study, as the experiment was designed to investigate the improvement of obesity and atherosclerosis by celastrol-enriched peanuts by controlling a single variable. The construction of the obesity and AS mouse models was uniformly based on a high-fat diet background, thus it was not possible to obtain the independent effect of celastrol in peanuts enriched with celastrol.
Comments 10: It the abstract the authors write “The findings indicate that cel-peanut significantly reduced serum levels of triglycerides (TG), total cholesterol (TC), and low-density lipoprotein cholesterol (LDL-C), while increasing high-density lipoprotein cholesterol (HDL-C).” however, this is no significant changes in LDL, HDL.
Response 10: Thanks for this suggestion.I apologize for the incorrect statement; TC and LDL did not show significant changes. I will modify my expression in the text:
Additionally, the serum TC levels and LDL-C exhibited a reduction trend in the HFD-Cel-peanut group mice, but there is no significant difference. (Line 338-339)
Reviewer 3 Report
Comments and Suggestions for Authors
The evaluation of the manuscript entitled “Effects of celastrol-enriched peanuts on metabolic health and the development of atherosclerosis ” by Yongting Luo and Peng An (corresponding authors) et al. sent to Nutrients.
The manuscript requires some improvements and explanations. In the present form, I think the description of the results and then subsequently in discussion section is a mixture of justified and unjustified claims (see comments). Overall, the text could provide some new light on the subject, but not in the present form.
Comments:
The authors used the term expression in relation to IL-6 and TNFalpha. I think they are wrong in using that. For instance about IL-6. IL-6 Expression; Definition: Refers to the production or synthesis of IL-6 (Interleukin-6), which is a cytokine involved in immune response. IL-6 expression typically refers to how much of the IL-6 gene is being actively transcribed into messenger RNA (mRNA) and then translated into the IL-6 protein by cells. Measurement: IL-6 expression is often measured at the level of gene transcription (e.g., using PCR or RNA sequencing) or protein synthesis (e.g., using Western blot or flow cytometry to look at the presence of IL-6 in cells).
IL-6 Level; Definition: Refers to the concentration or amount of IL-6 present in a biological sample (e.g., blood, serum, tissue fluid) at a given time. It reflects how much IL-6 is circulating in the body or present in a specific area, and is often a direct measurement of the protein itself. Measurement: IL-6 levels are typically quantified using techniques such as enzyme-linked immunosorbent assay (ELISA), immunoassays, or mass spectrometry.
The authors stated that “ Our results revealed that cel-peanut significantly reduced serum levels of triglycerides (TG), total cholesterol (TC), and low-density lipoprotein cholesterol (LDL-C), while increasing high-density lipoprotein cholesterol (HDL-C). Concurrently, cel-peanut markedly decreased atherosclerotic lesion area and enhanced collagen content within plaques. Mechanistic investigations demonstrated that cel-peanut reduced serum malondialdehyde (MDA) levels and suppressed the expression of pro-inflammatory cytokines (Tnf-α and Il-6) in atherosclerotic lesions”. Only the statement for TG is true. In the case of TC the differences between groups are P>0.05 (not significant); LDL the same (not significant); HDL a significant difference is between both peanut groups but not vs. control group; MDA, there is no difference vs. control group; TNF alpha not significant, and in the case of IL-6 control>cel-peanut but not in other cases. So, the conclusions driven by the authors are not justified.
In the Introduction section the authors stated that “Several signaling pathways like NLRP3 inflammation, toll-like receptors, lipoprotein conversant subtitle/kelvin type 9, Notch and Wnt signaling pathways are important for atherosclerosis development and regression”. But the authors did not measure those parameters. Why?
The authors stated in the Introduction: “celastrol can have cytotoxic, hepatocyte and even neurotic effects at high concentrations or in the case of prolonged exposure”. If they experimented with a novel celastrol form, they should show the benefit in relation to the liver at least.
Avertin is the trade name for a sedative and anesthetic compound known as tribromoethanol. Please do not use a trade name.
How the blood samples were collected?
Statistics: what post-hoc test was used after ANOVA analysis?
Line 304-305: “These results suggested that celastrol-enriched peanut could affect the body fat and lean mass percentage, and play a role in reducing fat and improving lean mass percentage.” Such sentences should not appear in the Result section but in the discussion. Please check the Results.
Line 317: Improvement in TC and LDL? There were NS differences. If you discuss the statistical trend, you must provide a P value. (line 598 too).
Line 320: HDL reduction? You mean LDL?
Figure 3: Please mark the most important histological elements on the images with arrows.
Line 667: Furthermore, this indicates that peanuts may act as an effective carrier for celastrol, enhancing its absorption in the digestive tract. (??) Did you measure the celastrol content in circulating blood? NO!. Did you compare supplementation with pure celastrol ? No. So, you cannot write that. Your future proposal is only proposal. Please conclude the results of the experiment conducted here.
Author Response
Comments 1: The authors used the term expression in relation to IL-6 and TNF-alpha. I think they are wrong in using that. For instance about IL-6. IL-6 Expression; Definition: Refers to the production or synthesis of IL-6 (Interleukin-6), which is a cytokine involved in immune response. IL-6 expression typically refers to how much of the IL-6 gene is being actively transcribed into messenger RNA (mRNA) and then translated into the IL-6 protein by cells. Measurement: IL-6 expression is often measured at the level of gene transcription (e.g., using PCR or RNA sequencing) or protein synthesis (e.g., using Western blot or flow cytometry to look at the presence of IL-6 in cells).IL-6 Level; Definition: Refers to the concentration or amount of IL-6 present in a biological sample (e.g., blood, serum, tissue fluid) at a given time. It reflects how much IL-6 is circulating in the body or present in a specific area, and is often a direct measurement of the protein itself. Measurement: IL-6 levels are typically quantified using techniques such as enzyme-linked immunosorbent assay (ELISA), immunoassays, or mass spectrometry.
Response 1: Thanks for this suggestion. I should have stated the concentration of IL-6 and TNF-α rather than expression, I will correct this in the article. (Line 28, 340, 347)
Comments 2: The authors stated that “Our results revealed that cel-peanut significantly reduced serum levels of triglycerides (TG), total cholesterol (TC), and low-density lipoprotein cholesterol (LDL-C), while increasing high-density lipoprotein cholesterol (HDL-C). Concurrently, cel-peanut markedly decreased atherosclerotic lesion area and enhanced collagen content within plaques. Mechanistic investigations demonstrated that cel-peanut reduced serum malondialdehyde (MDA) levels and suppressed the expression of pro-inflammatory cytokines (Tnf-α and Il-6) in atherosclerotic lesions”. Only the statement for TG is true. In the case of TC the differences between groups are P>0.05 (not significant); LDL the same (not significant); HDL a significant difference is between both peanut groups but not vs. control group; MDA, there is no difference vs. control group; TNF alpha not significant, and in the case of IL-6 control>cel-peanut but not in other cases. So, the conclusions driven by the authors are not justified.
Response 2: Thanks for this suggestion. I apologize for the incorrect statemen and I will modify my expression in the text. (Line 23-25)
Comments 3: In the Introduction section the authors stated that “Several signaling pathways like NLRP3 inflammation, toll-like receptors, lipoprotein conversant subtitle/kelvin type 9, Notch and Wnt signaling pathways are important for atherosclerosis development and regression”. But the authors did not measure those parameters. Why?
Response 3: Thanks for this question. The mechanism by which celastrol combats inflammation is currently unclear. Based on past experience, a broad spectrum of representative inflammatory indicators was measured. Since we are not clear about which specific inflammatory factors are at play, I will delete this sentence.
Comments 4: The authors stated in the Introduction: “celastrol can have cytotoxic, hepatocyte and even neurotic effects at high concentrations or in the case of prolonged exposure”. If they experimented with a novel celastrol form, they should show the benefit in relation to the liver at least.
Response 4: Thanks for this suggestion. This is a limitation of the present study, as the experiment did not include a group that was solely administered celastrol, therefore, it is not possible to obtain a direct comparison of the toxicity improvement effects between celastrol using plant carrier enrichment form and free celastrol form.
Comments 5: Avertin is the trade name for a sedative and anesthetic compound known as tribromoethanol. Please do not use a trade name.
Response 5: Thanks for this suggestion. I will change Avertin to Tribromoethyl alcohol. (Line 131)
Comments 6: How the blood samples were collected?
Response 6: Whole blood was obtained via cardiac puncture. (Line 132)
Comments 7: Statistics: what post-hoc test was used after ANOVA analysis?
Response 7: A Tukey multiple test was used after one-way ANOVA. (Line 282, 288, 330, 357,384, 490)
Comments 8: Line 304-305: “These results suggested that celastrol-enriched peanut could affect the body fat and lean mass percentage, and play a role in reducing fat and improving lean mass percentage.” Such sentences should not appear in the Result section but in the discussion. Please check the Results.
Response 8: Thanks for this suggestion. I will place this sentence in the discussion section. (Line 617-619)
Comments 9: Line 317: Improvement in TC and LDL? There were NS differences. If you discuss the statistical trend, you must provide a P value. (line 598 too).
Response 9: Thanks for this suggestion. There is no significant difference between TC and LDL, no p-value, there is a problem with my expression, I will change my expression. (Line 338)
Comments 10: Line 320: HDL reduction? You mean LDL?
Response 10: Yes, it is LDL, thank you for this suggestion, I will change it to LDL. (Line 337)
Comments 11: Figure 3: Please mark the most important histological elements on the images with arrows.
Response 11: Thanks for this suggestion. I will indicate with an arrow in Figure 3.
Comments 12: Line 667: Furthermore, this indicates that peanuts may act as an effective carrier for celastrol, enhancing its absorption in the digestive tract. (??) Did you measure the celastrol content in circulating blood? NO!. Did you compare supplementation with pure celastrol ? No. So, you cannot write that. Your future proposal is only proposal. Please conclude the results of the experiment conducted here.
Response 12: Thanks for this suggestion. I will put it into the future proposal, rather than writing it here, I will delete this sentence. (Line 677-678)
Round 2
Reviewer 2 Report
Comments and Suggestions for Authors
The authors addressed all my concerns.
Reviewer 3 Report
Comments and Suggestions for Authors
The revision is OK. Only one remark: under the Figure 3, you must indicate what is the "arrow" for?